# An Engaged Community of Faith to Decrease HIV Stigma in the U.S. South

**DOI:** 10.3390/ijerph20032100

**Published:** 2023-01-24

**Authors:** Latrice C. Pichon, Erin N. Jewell, Andrea Williams Stubbs, DeMarcus Jones, Bettina Campbell, Katrina M. Kimble, Gina M. Stewart, LaRonia Hurd-Sawyer, Lacretia Carroll, Terrinieka W. Powell

**Affiliations:** 1Division of Social and Behavioral Sciences, School of Public Health, University of Memphis, Memphis, TN 38152, USA; 2Headliners Memphis, Memphis, TN 38104, USA; 3Hill-Hernando Baptist Church, Hernando, MS 38632, USA; 4Brown Baptist Church, Southaven, MS 38671, USA; 5Christ Missionary Baptist Church, Memphis, TN 38106, USA; 6Partnership to End AIDS Status (PEAS), Inc., Memphis, TN 38115, USA; 7University of Tennessee Health Science Center, Memphis, TN 38163, USA; 8Department of Population, Family, and Reproductive Health, Johns Hopkins University Bloomberg School of Public Health, Baltimore, MD 21205, USA

**Keywords:** faith, HIV stigma, community engagement, CBPR, virtual conference, pandemic, prevention

## Abstract

Community members from a city in the U.S. Deep South identified root causes of HIV racial disparities, including stigma. This meeting report describes how we developed and implemented a conference series to address HIV stigma. We used community feedback and bidirectional learning to host two meetings in observance of National HIV Testing Day (June 2021) and National Southern HIV/AIDS Awareness Day (August 2021). We established a 10-member organizing committee workgroup that met monthly to plan the Faith Summit in honor of National Black HIV Awareness Day (February 2022). Lessons learned include (a) the effectiveness of different community engagement strategies, including participatory evaluative approaches, and (b) strategies to maintain engagement and increase participation, such as reliance on personal and professional networks and prompting the community about forthcoming interventions. Sustaining a conference series to end HIV stigma requires commitment and inclusive participation. This collaborative project offers additional evidence that faith communities can be a part of the solution to ending the HIV epidemic and related health disparities.

## 1. Introduction

Since the beginning of the HIV pandemic, faith communities have had an integral role in education, outreach, prevention, and care [1,2,3,4,5,6]. There is a growing body of research documenting the faith community’s involvement in HIV testing and behavioral interventions as well as stigma reduction [7]. However, the need to engage worship spaces and their congregations actively and efficiently in HIV persists. This is especially true in the U.S. Deep South, where researchers and practitioners continue to face challenges when addressing sexual health topics.

The Deep South is often defined as a subset of nine states, including Tennessee (project site). Situated in the complex Bible belt of the Southern U.S., conservative beliefs are deeply rooted in the culture and influence how prevention services are rendered. In the southern region, the HIV prevalence is 453 per 100,000 [8]. Providers often struggle with how best to care for members of the LGBTQ+ community due to their own personal beliefs regarding sexuality [9,10]. Federally qualified agencies are often faith-based in the Deep South and, in many cases, the sole source of care for people living with HIV. This leaves marginalized communities with limited options for prevention and quality care.

Because of the high density of churches in the Deep South, faith communities, comprised of community members, the congregation, and church leaders, must be included in efforts to end the HIV epidemic; they are a part of the solution. Practitioners in the Deep South often are also faith community members and come to public health with a set of attitudes and beliefs. Furthermore, Black men who have sex with men (MSM) and Black transwomen have a strong desire for spiritual connection [11,12,13,14]. Many people from these marginalized groups have grown up in traditional Black churches and struggle with acceptance from other church members [15,16]. They have suggested that faith communities can offer more inclusive HIV prevention efforts by (1) reducing homosexuality stigma by increasing interpersonal and institutional acceptance, and (2) addressing the sexual health needs of all congregants by offering universal and targeted sexual health promotion [17].

Stigma is one of the root causes of persistent health disparities among people living with HIV (PLWH) who also may be members of the LGBTQ+ community. HIV stigma in the South, often rooted in religious beliefs, influences the quality and quantity of care that LGBTQ+ individuals receive. While there is evidence that faith, religion, and spirituality may perpetuate stigma, partnering with local HIV service organizations to address HIV stigma within faith communities may be one solution for reducing such stigma [18,19]. Several community-based organizations, LGBTQ+ community members, and allies attend local churches and worship services. 

Previous research has found that HIV stigma and education interventions are successful when connecting with faith communities. Twelve churches in rural Alabama enrolled in a pilot HIV anti-stigma intervention. Groups were assessed using pre- and post-intervention surveys which found that the intervention group reported significant reductions in individual-level stigma compared with the control group (*p* = 0.02) [6]. Similarly, two Black churches in southern California enrolled in an HIV intervention measuring stigma and mistrust at the individual, congregational, and community levels. Findings indicated individuals who have negative attitudes toward drug addiction and homosexuality were associated with higher levels of stigma, but in contrast, knowing someone living with HIV combats those negative attitudes [20].

Formative work in assessing organizational readiness was conducted to assess the needs among faith leaders in the Deep South to engage in HIV prevention and education. We found that securing financial and human resources facilitates readiness [21]. Subsequently, we applied for and were awarded a federal grant in collaboration with two community partners and the university to plan a conference series to engage faith leaders in a discussion on the role of stigma in HIV prevention and care and reassess church readiness in addressing HIV-related stigma in faith-based settings.

This paper describes our community-engaged strategies to host planning meetings and an eventual faith summit to “Fight Stigma with Faith”. During these virtual community events, we sought to develop and implement potential solutions to address HIV disparities and ultimately reduce HIV stigma. We provided education in two community planning meetings and shared findings from a stigma reduction photovoice project. Finally, we worked toward identifying a spectrum of actions that faith leaders could choose from given their varying degrees of readiness to adapt their ministries to this cause, and in developing strategies for ushering faith leaders to higher levels of commitment over time.

## 2. Materials and Methods

### 2.1. Setting

Memphis, TN, is the home of rhythm and blues, barbeque, and the civil rights movement. There is a rich history of cultural influence on social issues driving health inequities. In 2019, the Memphis metropolitan area was identified as 1 of 48 “hotspots” for the growing number of new HIV infections. Memphis ranks 3rd among all U.S. metro areas for new HIV infections with a prevalence of 537.9 compared with the national average of 321.9 [22]. Memphis HIV rates rank third due to several factors impacting the social determinants of health. One of the main factors unique to the South, especially Memphis, is the stigma of HIV due to many conservative religious beliefs. This leads to little discussion of the risk factors, preventative measures, and treatment options available regarding HIV. The community worked together to devise a community action plan to address root causes including HIV stigma [23]. An identified strategy to address stigma was to engage the local faith community. There are more than 2000 churches in the city of Memphis. We used a virtual platform (Zoom) to keep participants COVID-19 safe and to maintain consistent engagement given infection rates during the time of the Faith Summit.

### 2.2. CBPR Partnerships and Organizations

Monthly faith organizing committee meetings included the three members of the core investigative team (Pichon, Campbell, and Powell) and ten community members with interest and/or experience in the faith community and/or HIV education from our partner organizations. Via bi-directional learning approaches, we presented basic information allowing time for open discussion and feedback using PowerPoint presentations, topic rankings, and consensus voting via Qualtrics surveys to inform the content of the virtual conference.

Throughout the course of this project, we partnered with Headliners Memphis to address internalized stigma and stigma reduction within faith communities. Headliners Memphis is a grassroots organization created to empower Memphis LGBTQ+ communities of color through HIV awareness, advocacy, entertainment, mobilization, and education [24,25]. For the past seven years, Headliners Memphis has served as a resource hub for queer communities of color due to our influence and connectivity within the Memphis area. An area of interest for Headliners Memphis is connecting with the faith-based community. However, the members of Headliners Memphis have noted that few local churches address HIV or the stigmatization of the disease within its institutions. Through hosting and sponsoring several underground community projects in non-traditional spaces, Headliners Memphis has successfully engaged MSM and the transgender community by providing on-demand testing, promotion of PrEP, and linkage to care for PLWH [26].

Our Memphis community HIV advisory board, Connect to Protect Memphis (C2P), has always had representation from the faith community on the 25-member coalition. Early in the conception of C2P, faith communities were identified as an essential potential partner in HIV education and prevention. Yet, membership and efforts ebbed and flowed over time. With the onset of the National Ending the HIV Epidemic (EHE) initiative [27], our local community re-engaged the faith community in the following ways [23]: addressing modifiable risk factors such as knowing HIV status and increasing testing uptake; developing system and organizational level policies to introduce church programming; and ultimately eliminating HIV disparities via prevention and access to quality care.

### 2.3. Procedures

In partnership with Headliners Memphis and Connect 2 Protect Memphis, we applied for and were awarded an NIH NIMHD R13 conference series grant. We hosted two virtual planning meetings in recognition of National HIV observance days. In June 2021, we hosted the first planning meeting in observance of National HIV Testing Day. The objective of this meeting was to commemorate the 40th anniversary of the first reported AIDS case with an overview of the importance of calling faith leaders to action and encouraging testing. In August 2021, we hosted the second planning meeting in observance of Southern HIV/AIDS Awareness Day. The objective was to bring awareness to the disproportionate impact of HIV in the South and to highlight previous community–faith–academic partnership successes applying bi-directional learning practices and shared decision making. Both planning meetings used photovoice storyboards from the Snap Out Stigma project [28]. This project, originating in 2019, provided 35 participants currently living with HIV in Memphis with cameras and allowed them to reflect on the lived experiences of internalizations of stigma through pictures, interviews, and focus groups. The information captured was then used to create storyboards for each participant which described their HIV experience.

From September 2021 through December 2021, the organizing committee met monthly and bi-weekly in January and February 2022 to finalize the number of sessions, session topics, speakers, branding, and other meeting logistics. The virtual faith summit “Fight Stigma with Faith” was hosted in February 2022 in observance of National Black HIV Awareness Day. The summit topics included the social determinants of HIV, HIV prevention, and HIV programs for faith communities. Presenters included senior pastors from two Mid-South churches, an HIV clinician, a former Ryan White Program manager, an HIV ministry leader with a social work background, and a public health academic researcher. We also issued Red Ribbon Awards (e.g., faith leadership, public health, and community advocacy) named after accomplished members of the HIV research and outreach community to three recipients.

### 2.4. Participatory Recruitment

The organizing committee promoted the main event via personal and organizational social media websites (e.g., Facebook and LinkedIn), listservs, and other media. A complete listing is detailed in the Acknowledgements Section.

### 2.5. Participatory Evaluation

We employed several methods to collect evaluative data for the Faith Summit including a pre-conference assessment (n = 113), Zoom polls during two planning meetings (n = 41), as well as a post-conference follow-up evaluation (n = 39) and non-attendance survey (n = 7). We used the chat box feature to gather participant feedback and transcripts from all virtual sessions. Polling attendees during the conference was necessary to obtain immediate feedback on comprehension and/or interest.

## 3. Results

### 3.1. Planning the Faith Summit

A total of 73 faith attendees participated in the first two planning meetings (n = 25 planning mtg 1 and n = 48 planning mtg 2). The need for more education in the faith community to dispel HIV myths was an overarching theme emerging from the events. Major discussion points included: the (a) lack of knowledge on the availability and usage of PrEP; (b) lack of awareness of HIV treatment options; and (c) the need for more compassion and love instead of judgement. Next, we convened the inaugural faith community organizing committee, including attendees from the initial planning meetings and existing partners from ongoing partnerships with C2P and Headliners Memphis.

We held seven organizing committee meetings over six months (Table 1). This core group guided the process by setting goals, deciding deliverables, and establishing a feasible timeline to host the virtual faith summit successfully. Session topics were narrowed down from an original list of 20 to the top 12 by our organizing committee. The topics “Developing HIV Ministries, General HIV Information, Mental Health Needs, and Community Safe Spaces” held the highest number of votes and were combined/collapsed to the following three: (1) General HIV Information, (2) Impact of Poverty on HIV, and (3) Faith Community’s Response (Figure 1).

We identified prominent speakers from the Memphis community with expertise in the three session topic areas. We identified the emcee who led the community HIV coalition for 13 years to open the meeting and transition from speaker to speaker. The committee agreed to have a Black nurse practitioner who has cared for youth and young adults at St. Jude Children’s Research Hospital Infectious Diseases clinic to provide education on HIV testing and biomedical prevention such as PrEP. This nurse co-presented with our former Memphis Ryan White Program manager who is a lifetime parishioner of a well-respected non-denominational church serving Black communities especially in mental health. An international faith leader, with countless years of experience working with Black congregations worldwide, was chosen to discuss social determinants of HIV including poverty, housing, and education. Finally, two national experts in faith-based intervention development and implementation shared strategies to effectively work with congregations to offer culturally congruent programs to address HIV.

The faith organizing committee also spent considerable time on branding. Collectively, we worked with a graphic design consultant, who identifies as LGBTQ+, to create a project logo to reflect the goals of the faith summit. The committee wanted to be inclusive of all faiths and a cross and ribbon were blended into a heart with the tagline “Fight Stigma with Faith” (Figure 2). The color red was used as an expression of love for the heart and an acknowledgement of the red ribbon’s significance for HIV. We used this new branding to develop a faith summit flyer to attract faith communities to save the date and register for the meeting. Similarly, the committee worked with a local company to design the Red Ribbon Award glass plaques to honor Red Ribbon Heroes during the award ceremony.

### 3.2. Hosting the Faith Summit

We administered a pre-survey (n = 113) during the faith summit registration process to better tailor session content. The pre-survey included three questions assessing demographic information (Table 2) and seven questions that measured the current knowledge base of registrants and informed content of the Faith Summit. Less than a quarter of registrants reported their church or faith-based organization had an HIV ministry (20%) in the pre-conference survey. A third of registrants reported that their church/FBO never had HIV programming. We asked registrants to rate their knowledge of HIV pre-conference, 46% said they had "a little" to "some" knowledge. Almost 27% of registrants reported their pastor or other faith leader had delivered a sermon on HIV/AIDS in their church or place of worship. We asked registrants if they knew anyone living with HIV, and 83% reported they personally have known someone living with HIV/AIDS. Additionally, 54% of registrants had high interest in adding HIV/AIDS programming to their places of worship, while 34% had medium interest, and 12% had low interest. Finally, 26.6% of registrants knew their faith-based organization/church provided HIV testing. These data subsequently informed the conference e-booklet describing each speaker’s bio, the conference agenda, and session topics.

A total of 68 unique viewers attended the virtual faith summit. In the first session, 22% of attendees had yet to hear of PrEP before the faith summit. By the end of the session, 100% of the attendees reported that there were medical treatments available to help a person living with HIV live longer. At the start of the session on the social determinants of HIV, 24% believed they had some expertise in the subject matter, by the end of the session, the level of knowledge had increased to 48%. 

In the last session of the faith summit, we asked attendees about their interest level in adding programming to address HIV/AIDS in their church. Almost two-thirds (62%) of attendees endorsed a high level of interest. Five participants reported their desire to increase their level of participation and provided contact information for membership in the faith HIV education advisory committee workgroup. We also assessed attendee interest in hearing about resources from the faith advisory board. Overwhelmingly, 93.1% of attendees responded affirmatively "yes" that they would be interested in additional resources. With input from the committee, an MPH practicum student researched up-to-date contact information for mental health services, HIV testing and care, transportation, and other supportive community services resulting in the Mid-South Faith Summit Resource Guide that was disseminated to conference attendees the following National Southern HIV/AIDS Awareness Day in August 2022.

## 4. Discussion

### 4.1. Lessons Learned

#### 4.1.1. Build onto Existing Relationships to Enhance Engagement

We employed a three-prong approach to keep engaging our audience: (1) working with community members/partners we know; (2) inviting community members we do not know; and (3) inviting community members for future interventions. First, each member of the organizing committee contacted network members directly. Second, we compiled an email distribution listing and "cold called" churches. Finally, during the last session of the Faith Summit, the two presenters previewed the development and implementation of future interventions under the direction of the PI (CDC MARI grant 1 U01PS005211-01-00) to prompt attendees for the next steps.

Administering pre-and post-conference surveys helped the organizing committee to have an approximate head count for faith summit attendance as well as gather pre-existing participant HIV knowledge and understand future attendee needs post-conference. This assessment was not conducted to measure pre- and post-survey data as we quickly learned registrants were not always the same people as those who attended. In fact, registrants were more likely to be a "no show", versus those who attended were more likely not to have registered. We sent email blasts/reminders weeks before the faith summit, the week of, day before, and the day of. On the morning of the faith summit, the PI and Spiritual leader used social media platforms to prompt members of their personal and professional networks to join the live webinar. Figure 3 illustrates registration tracking over time.

#### 4.1.2. Use Tools to Make Virtual Meetings Interactive

Zoom polling proved to be an effective strategy to engage meeting attendees. It facilitated session topic transitions and allowed for needed speaker breaks. Interactive methods reduced Zoom fatigue and allowed the research team to collect evaluative data to assess the speaker’s effectiveness with structured survey polls. The Zoom chat box feature allowed the team to collect qualitative feedback from attendees as one participant stated: “Not a question; just a praise! appreciate the approach of grace. It is the Body’s job to love.” This option also allowed attendees to ask questions as one attendee asked: “Is PrEP available for minors in TN?”

Valdez and Gabruim discussed methods to maintain participatory research during COVID-19 through online platforms such as Zoom by facilitating group discussions through photovoice and digital storytelling [29]. We engaged faith community members in our two planning meetings for National HIV Awareness Day in June and National Southern HIV Awareness Day in August using examples from our photovoice storyboards. Disseminating photovoice findings through virtual platforms has been an effective method to continue CBPR and by informing content for the Faith Summit. We established a faith advisory committee to launch a faith summit in February 2022. Breny and McMorrow found that many participants often felt more comfortable and engaged more in the discussion via Zoom by using the chat feature [30]. Similarly, we used this feature as well as Zoom polls to increase accessibility. These features could be advantageous for data collection that is not available during in-person exhibitions. Valdez and Gabrium also mention the need for research members to continually generate notes on emerging themes throughout project implementation [29]. Following each Snap Out Stigma e-exhibit or virtual presentation, research team members developed summaries of lessons learned via field notes, participant feedback, and methods for improvement.

Over the nine months of planning the virtual conference series, we discovered a few things about people’s education and comfortability when it came to the topics of HIV and sexual health. First, the utility of person-centered language and the growing need to brainstorm ways to bridge the gap to younger generations were noted. Second, we observed the value of personal testimonials (i.e., “Faces of HIV”) in our virtual photovoice exhibit. Our hope was that, by sharing the lived experiences of PLWH (positive and negative), we would clarify the impacts of stigmatization and people would see the need for increased education and change. Thirdly, improving community engagement strategies as an ongoing goal of the partnership. These three findings are congruent with the existing literature assessing faith leaders’ comfort in implementing HIV programs [21,31].

Creating safe relational spaces for youth and young adults could be strengthened by working more closely together to identify the faith-based organizations and their faith leaders and build environments that feel safe to them [10]. A person can be a part of the HIV coalition or Headliners Memphis but not technically feel “safe” with other individuals outside of the network of the coalition and grassroots organizations. Youth and young adults may gravitate toward non-denominational churches that have a younger audience or are more progressive or gravitate to someone outside of that particular network [32]. When people feel safe, they tend to speak up, are more likely to participate, and have other community members participate as well [33]. A critical component in creating safe relational spaces is acknowledging that the faith summit and faith-based research efforts, in general, are not to challenge doctrine but to focus on healthy choices, healthy people, and reducing disease [34].

We recognize some topics such as sexuality to still be met with resistance [35,36]. Future work might consider employing best practices within the community to advance reconciliation and healing around this topic. Facilitating conversations about harm reduction, “agreeing to disagree”, and using circle/group discussions to promote healing are needed. Focusing on solution-oriented strategies such as scaling up the organizational capacity building, bi-directional training with church members and marginalized people on professionalism and allyship, and insurance of provision of a sense of safety outside community and empathy with the community should be integrated into restorative co-learning modules.

As we continue to refine community-engaged strategies and meet people where they are, using intergenerational practices to increase participation is paramount. Social media, specifically Instagram, TikTok, and YouTube, can complement the current platforms we use (e.g., Facebook and LinkedIn) to better engage multi-generations [37,38,39,40]. Many of the older individuals in Memphis prefer receiving letter mail and paper copies of educational resources. It may have been helpful for faith-based organizations to send mailers to their members to increase awareness and their participation in the faith summit and planning meetings.

The next steps involve growing our faith community advisory committee. We reached out to the five participants showing interest in future faith HIV education advisory committee service, three of whom have agreed to be on the newly established advisory team. We will share the YouTube link of the faith summit and other meeting recordings around future national observance days in hopes to reach more inclusive audiences. We will also design a health education program for Deep South churches integrating community input from the lessons learned from the planning meetings and faith summit.

#### 4.1.3. Limitations

There are several limitations associated with the Faith Summit. First, there is a dearth of prior research on implementing virtual conferences in this context of engaging faith communities. Literature on this type of virtual event is scarce, and therefore left space for error or missed opportunities; however, there is great opportunity to expand HIV education and networking with communities of faith via virtual platforms. Second, we limited marketing of the Faith Summit through virtual strategies such as professional email, e-listservs, and social media. In the future, to increase engagement we intend to print the conference flyer to display in faith spaces or use USPS mail services to send flyers to our list of churches in the MSA. This could have been accomplished with more time allotted between the development and finalization of the logo and flyer and the conference date. Another study limitation is the number of conference registrants compared with those who actually attended the conference. The attendee list was not congruent with the registration list. There were some conference attendees who did not register and several who registered but did not attend. Due to this gap, we were not able to collect pre-survey data for those who attended but did not register.

## 5. Conclusions

Engaging the faith community to address HIV stigma is well-aligned with local EHE implementation goals. The decision to host a virtual summit given the global COVID-19 pandemic and local safety protocols was acceptable and feasible with human and technological resources. Areas for future growth include: (1) reaching more members of the LGBTQ+ faith community or a larger presence of affirming churches; (2) involving more members of the LGBTQ+ community in conference organizing leadership: and (3) garnering support from latent churches who fall on the low readiness continuum.

## Figures and Tables

**Figure 1 ijerph-20-02100-f001:**
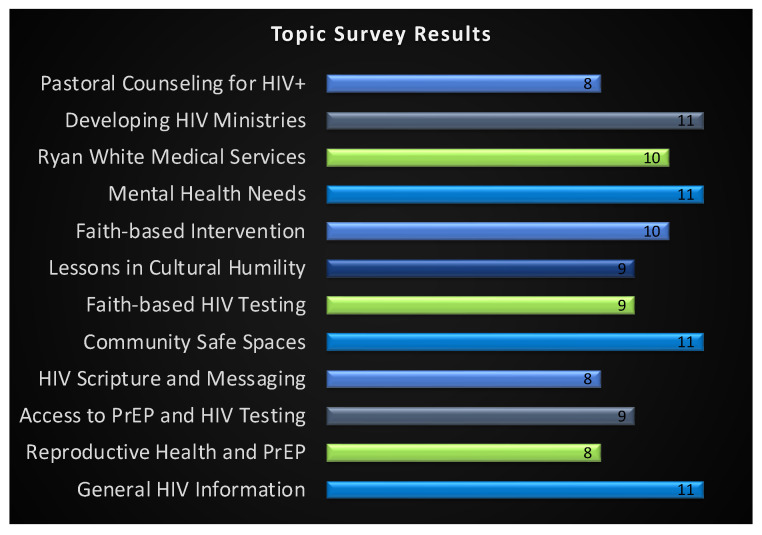
Faith Summit Topic Rankings (n = 12 responses recorded). There was a tie for 12 topics with the most votes from the survey. Topics were integrated into 1 session to condense (e.g., General HIV Information and PrEP and HIV testing). Data are displayed with total votes at the end of the data bars.

**Figure 2 ijerph-20-02100-f002:**
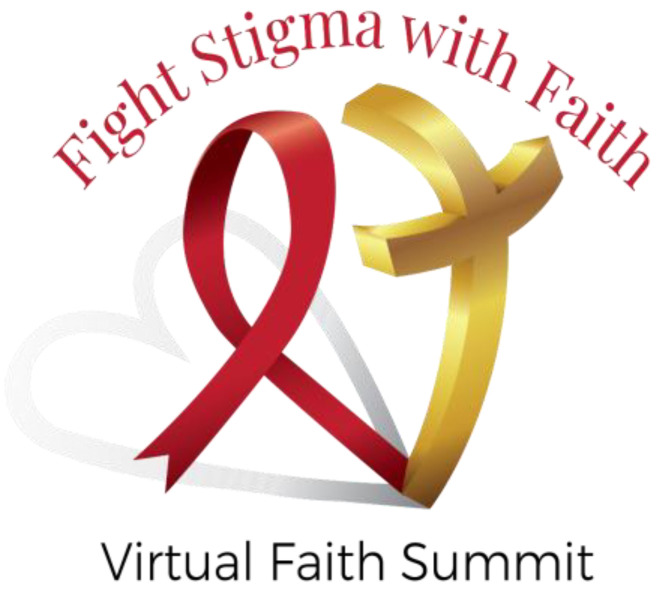
Project Logo.

**Figure 3 ijerph-20-02100-f003:**
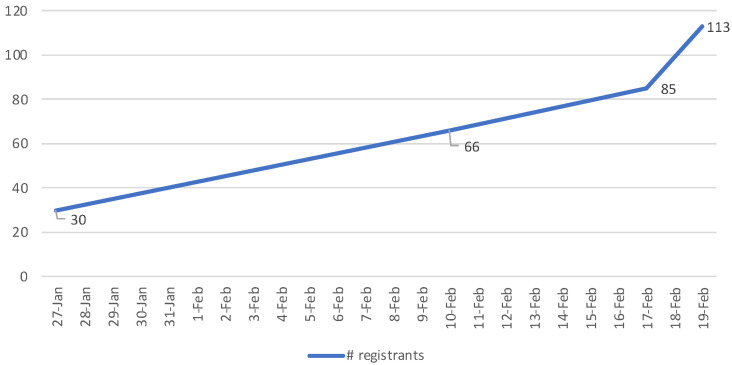
Faith Summit Registration Tracking.

**Table 1 ijerph-20-02100-t001:** Conference Organizing Committee Meeting Purpose and Outcomes.

Meeting Date	Agenda Purpose	Outcome
30 September 2021	Covered lessons learned from Planning Mtg 1 and Planning Mtg 2	Confirmed summit date and location.
28 October 2021	Presented survey results on topic rankings	Agreed on General HIV Information, Impact of Poverty on HIV, Faith Community’s Response as topics (Figure 1).
18 November 2021	Discussed conference pack	Finalized session times and order of presentation.
16 December 2021	Provided Faith Summit planning updates	Confirmed speakers. Finalized preregistration Qualtrics survey. Finalized nomination from Qualtrics.
13 January 2022	Brainstormed recruitment and advertisement strategies	Finalized listservs to send flyer. Confirmed creative consultant and prepared contract.
27 January 2022	Discussed Red Ribbon Award Nominations	Updated nomination status. Designed glass plaque award layout.
10 February 2022	Discussed poll questions Brainstormed presenter content	Developed conference evaluation survey.
17 February 2022	Introduced: Consultant responsible for run of show MPH student responsible for e-booklet MPH student responsible for resource guide development	Scheduled date for dress rehearsal. Finalized e-booklet (e.g., cover page, Table of Contents, Agenda, Session Topics, Speaker Bios, Acknowledgements). Brainstormed topics for resource guide. Updated recruitment tracker.

**Table 2 ijerph-20-02100-t002:** Key characteristics of Faith Summit registrants (n = 68).

Variable	N	%
Sex		
Male	14	20.6
Female	54	79.4
Faith Denomination		
Baptist	13	19.1
Non-denominational	7	10.3
Other	9	13.2
Did not disclose	39	52.4
Church Affiliation		
Yes	33	48.5
No/Did not disclose	35	51.5

## Data Availability

No new data were created and analyzed in this study. Data sharing is not applicable to this article.

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
