# Peer review of "An Engaged Community of Faith to Decrease HIV Stigma in the U.S. South"

_ijerph, 2023, doi:10.3390/ijerph20032100_

Round 1
Reviewer 1 Report
The report describes how the authors developed and implemented a conference to address HIV related stigma via faith based communities. While report findings have the potential to inform the field, specifically, focused on stigma reduction, there are a few weaknesses that need to be addressed.
Background:
1. Need to provide HIV prevalence [ranges] for the Deep South.
2. Briefly discuss what kinds of behavioral and stigma reduction interventions have been implemented by the faith community in the South, instead of directing the reader to other publications.
3. "Consistently, we partnered with Headliners Memphis to address internalized stigma and stigma reduction within faith communities." Please provide details on the specific strategies you have used used to address internalized stigma and overall stigma reduction, as well as your findings.
Materials and Methods
4. Under setting: Please provide the HIV prevalence rate of Memphis compared to the National/US average.
5. Also, a sentence or 2 on why Memphis ranks highly with new infections may be helpful for the reader to conceptualize, and to provide more justification for the setting.
6. Procedures: What is “Snap Out of Stigma”? Was this a stigma reduction project?? Please provide the relevant context.
Results
7. “Fight Stigma with Faith” is mis referenced as “fig 1” in the narrative on page 5.
8. How many questions were included in the pre-survey? The authors only report results from less than 5 questions. Please specify the information/ data obtained.
9. A total of 68 viewers attended the summit. Is there a way to at least understand the make-up pf participants? I am assuming this information was captured on the survey?? This information is helpful to understand your attendees and survey respondents.
10. On page 7, the authors, again, refer to their photovoice project but no details are provided on this project. I would strongly recommend providing a brief description of this study on the first mention in the background or methods section. Same for "Snap out of Stigma” on page 8.
11. Adding a section on limitations, which are several, will be helpful.
Author Response
Please see the newest attachment.

Reviewer 2 Report
Please see the attachment.

Author Response
Please see the newest attachment.
